# Classification of Fresh and Frozen-Thawed Beef Using a Hyperspectral Imaging Sensor and Machine Learning

Seongmin Park [1,2], Suk-Ju Hong [3], Sungjay Kim [1], Jiwon Ryu [1,3], Seungwoo Roh [1,3] and Ghiseok Kim [1,2,3,*]

1 Department of Biosystems Engineering, College of Agriculture and Life Sciences, Seoul National University, 1 Gwanak-ro, Gwanak-gu, Seoul 08826, Republic of Korea; wildhammer@snu.ac.kr (S.P.)
2 Research Institute of Agriculture and Life Sciences, Seoul National University, 1 Gwanak-ro, Gwanak-gu, Seoul 08826, Republic of Korea
3 Global Smart Farm Convergence Major, College of Agriculture and Life Sciences, Seoul National University, 1 Gwanak-ro, Gwanak-gu, Seoul 08826, Republic of Korea; hsj5596@snu.ac.kr
* Correspondence: ghiseok@snu.ac.kr

**Abstract:** The demand for safe and edible meat has led to the advancement of freeze-storage techniques, but falsely labeled thawed meat remains an issue. Many methods have been proposed for this purpose, but they all destroy the sample and can only be performed in the laboratory by skilled personnel. In this study, hyperspectral image data were used to construct a machine learning (ML) model to discriminate between freshly refrigerated, long-term refrigerated, and thawed beef meat samples. With four pre-processing methods, a total of five datasets were prepared to construct an ML model. The PLS-DA and SVM techniques were used to construct the models, and the performance was highest for the SVM model applying scatter correction and the RBF kernel function. These results suggest that it is possible to construct a prediction model to distinguish between fresh and non-fresh meat using the spectra obtained by purifying hyperspectral image data cubes, which can be a rapid and non-invasive method for routine analyses of the meat storage state.

**Keywords:** beef; drip loss; hyperspectral imaging; PLS-DA; support vector machine

## 1. Introduction

With the increased meat consumption in modern society, the demand for hygienically safe and edible meat has continuously increased [1]. Notably, meat is more prone to decomposition than other food products, and this has led to the advancement of freeze storage techniques [2]. Freezing storage is an effective means of preservation against microbial contamination of meat, but the quality and sensory properties of meat generally decrease after freeze storage [3,4]. This is because the cell membranes in muscle tissues are damaged by ice crystals during freezing and subsequent thawing, whereby pro-oxidants, such as heme iron, are released to accelerate lipid oxidation [5]. Given this situation, suppliers sometimes make unjust profits by falsely labeling meat that has been freeze-stored as fresh meat [6,7].

The most common method to distinguish between fresh meat and thawed meat used by researchers is to measure the activity of an enzyme called β-hydroxyacyl-CoA-dehydrogenase [8]. The technique was suggested for the first time by Gottesmann and Hamm [9], and its basic principle is the release of glycolytic enzymes to the cytosol as the mitochondria inside cells are destroyed in the process of meat freezing and thawing. However, with the advancement of freeze-storage techniques, different methods, such as slow freezing, air-blast freezing, and cryogenic freezing have been developed, and studies have shown that the use of these special methods can prevent enzyme release [10]. This indicates that although these freezing methods can prevent the deterioration in meat quality,

the discrimination of fresh meat from thawed meat by measuring the enzyme activities cannot be easily achieved.

Therefore, a group of researchers attempted to solve the problem of identifying thawed meat through the use of nuclear magnetic resonance (NMR), a method that analyzes the energy absorption by atomic nuclei based on the spin applied in a magnetic field [11,12]. The presence of a nucleus around a molecule could affect the energy absorption by atomic nuclei and create small changes in the external magnetic field. Through this process, comprehensive data on food molecular structures can be obtained. Mortensen [13] measured the transverse and longitudinal relaxation time in a magnetic field for pork and lamb meats to verify the changes caused by the process of freezing and thawing. In addition, LF-NMR is claimed to be a powerful method for monitoring proton migration and identifying the distribution of water components during food processing [14]. However, despite the advantage that NMR is a universal method, the cost of measurement and operation are considerably high and only a trained technician can perform the measurements [15].

For consumers to determine a quality deterioration or the thawed state of beef at the time of purchase, they rely solely on the observations of the meat color, aroma, and the date of slaughter [16]. In a laboratory environment, chemical (pH, volatile basic nitrogen and thiobarbituric acid reactive substances) or microbiological (total aerobic bacterial count) methods are conventionally used to estimate meat freshness, and physical methods (meat color, shear force, and water holding capacity) are used to measure meat texture after cooking [17,18]. However, these methods uniformly require the meat sample to be destroyed or cooked and the time taken between the measurement and the outcome is long; besides, there is a possibility of introducing errors depending on the skillfulness of the technician. Therefore, a technique to enable simple as well as rapid measurements of the past storage and current state of meat is expected to be useful in quality assurance and inspection processes when independently applied by large-scale manufacturers [19].

Near-infrared (NIR) spectroscopy has attracted the interest of many researchers because it offers more definite benefits compared with the aforementioned techniques [20,21]. The increased consumer interest in food quality and safety has necessitated the rapid monitoring of the molecular structure and chemical components of food. NIR spectroscopy instantaneously produces the spectra of liquid or solid samples without preprocessing, which could be used to predict the chemical and physical parameters [22]. Such functions are especially suited to simple, rapid, and non-invasive characterization of samples.

In a recent study, multi-spectral imaging (MSI) and Fourier transform infrared spectroscopy (FT-IR) were used to rapidly detect the thawed state of minced beef [23]. The researchers grouped the minced beef samples into 'immediately after processing', '7 days after freezing' and '32 days after freezing' and collected the respective MSI and FT-IR data. The resulting 105 datasets were analyzed using a partial least-squares discriminant analysis (PLS-DA) and support vector machine (SVM). The classification accuracy of fresh meat was 100% for MSI data, while it was 93.3% in PLS-DA and 86.7% in SVM for FT-IR data.

Research into hyperspectral imaging began with the observation that conventional NIR spectroscopy offers only a single spectrum per wavelength band without spatial information [24,25]. NIR hyperspectral images can be used to visualize the components in measured samples, giving them an advantage over NIR spectral images. A study was conducted to classify pork, beef and lamb meat using hyperspectral image data, principal component analysis (PCA) and PLS-DA [26]. In spectrum-based models, six optimal wavelengths (957, 1071, 1121, 1144, 1368, and 1394 nm) were selected as the criteria for identification, and the reported classification accuracy was 98.67%. In a follow-up study by the same researchers, a hyperspectral imaging system of the Vis-NIR domain was suggested. The optimal wavelengths to detect horse meat mixed with minced beef were 515, 595, 650, and 880 nm, and when the result was applied in the construction of a partial-least-squares regression (PLSR) model, the estimated $R^2$ was 0.98 [27]. However, these studies were still limited in terms of analyzing the changes in the state of an identical sample.

Modern consumers are currently more educated than before with regard to the effects of meat and processed meat products on their health, which underlies the consumer demand for more transparent disclosure of information on food production processes [28]. Therefore, it is necessary to develop a rapid and non-invasive method for routine analyses [29]. NIR spectral data are accurate, reliable, and reproducible; hence, hyperspectral image data were used in this study to construct a machine learning (ML) model to discriminate between freshly refrigerated, long-term refrigerated, and thawed beef meat samples from identical carcasses. The highlights of this study are as follows: (1) NIR hyperspectral image data were used to construct a model to identify the thawed state of meat and classify the meat storage state without applying an invasive procedure on beef samples; (2) the PLS-DA technique and the SVM technique with kernel functions performed almost equally in developing a model for determining the status of meat through spectra; and (3) the approach is relatively rapid and non-destructive, allowing for routine analyses of the meat storage state, making it an attractive option for quality control in the meat industry.

## 2. Materials and Methods

### 2.1. Test Material and Experimental Classes

The test material in this study was the *M. longissimus thoracis* part of beef produced in South Korea. Only the *M. longissimus thoracis* part was cut from three different beef carcasses, vacuum-packaged, and transferred to the laboratory. At the laboratory, the vacuum packaging, the fascia, and surface fat were removed, and 81 samples were prepared, each containing $250 \pm 20$ g of isolated and aspiration-packed meat. The samples were cut immediately before the experiment; approximately 200 g was used in drip loss experiments and approximately 50 g was used for collecting the hyperspectral image data. The samples were divided into three experimental classes to be subsequently described. Each class contained an equal proportion of the three beef carcasses.

As shown in Figure 1, the refrigeration and freezing protocol was applied to the samples divided into three classes ('Fresh', 'Abused', and 'Frozen'). The 'Fresh meat' class was defined as the refrigerated meat samples suitable for consumption. To retain the fresh state, two groups of 'Fresh meat' class were refrigerated for 12 h and 24 h, respectively, after removing the vacuum packaging, and then both hyperspectral images and drip loss values were collected. The 'Abused meat' class was the refrigerated meat samples with possible quality deterioration caused by long-term refrigeration without freezing. These samples were refrigerated for 7 and 14 d, respectively, and hyperspectral images and drip loss were collected on those days. Additionally, the 'Frozen meat' class was frozen at $-20 \pm 2$ °C for 6 d, after which they were thawed for 24 h at $23 \pm 1$ °C, and hyperspectral images were collected and drip loss experiments were conducted. The refrigeration temperature applied to the 'Fresh meat' class and the 'Abused meat' class was $4 \pm 1$ °C using a constant temperature and humidity chamber (DS-543-CO, Dasol Scientific Co., Ltd., Hwaseong, Republic of Korea).

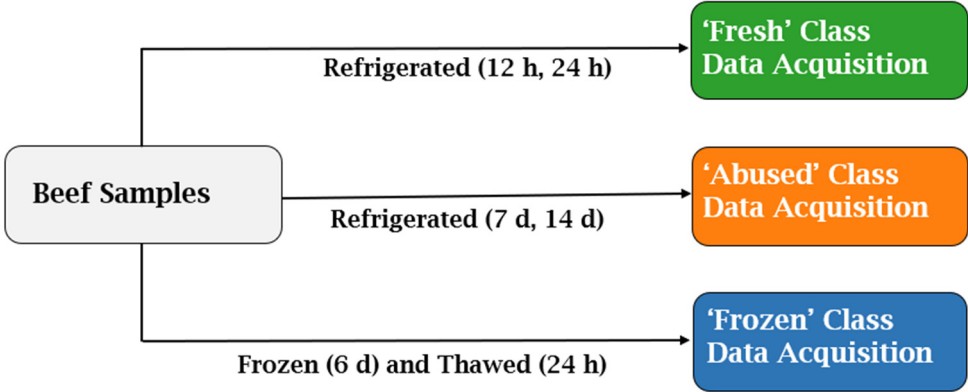

**Figure 1.** Refrigeration and freezing protocol applied to the samples divided into three classes in this study.

## 2.2. Drip Loss Test

Drip loss experiments were performed to confirm that the difference between each experimental group was significant. A total of 81 samples, 27 from each of the experimental classes, were analyzed. The tests were performed in the order described below, with a few adjustments that were based on previous studies [30,31]. For the test, the samples were cut into cubes of $20 \times 20 \times 20$ mm and placed in polyethylene bags. The experiment was performed by storing the samples at 4 °C for 6 d and measuring the drip loss by weighing them before and after being stored. The estimated drip loss was determined by the following equation (Equation (1)):

$$\text{Drip loss } (\%) = \frac{A - B}{A} \times 100, \tag{1}$$

where A is the weight of the sample measured before storage and B is the weight after storage.

## 2.3. Data Acquisition and Image Processing

In this study, an image acquisition system was installed in a dark room to ensure the complete elimination of external light and to enable the acquisition of hyperspectral images. An illustration of the entire system inside the darkroom is shown in Figure 2. The hyperspectral camera used in this study was a Pika L (Resonon Inc., Bozeman, MT, USA), which allowed the acquisition of hyperspectral image data cubes in the 400–1000 nm band of the NIR spectrum. To facilitate spectral acquisition, a stepper motor was incorporated into the system to move the camera directly along the horizontal direction. SoLux 4700 K MR-16 (Tailored Lighting Inc., Rochester, NY, USA) tungsten halogen lamps were used for sample illumination to ensure uniform illumination throughout the imaging process. Spectronon Pro 2.134 (Resonon Inc.) software was used to store the hyperspectral image data acquired by the camera.

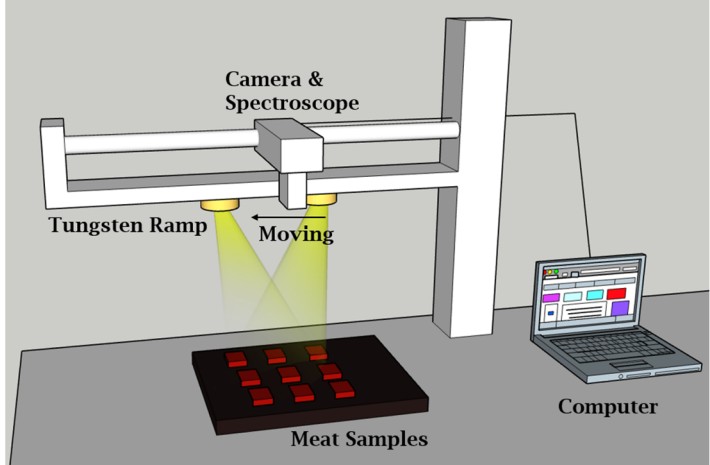

**Figure 2.** Schematic configuration of the hyperspectral data acquisition system installed in a darkroom.

For the purpose of this study, nine samples of the same experimental class were placed on a matte black plate at the same time and hyperspectral image data cubes were obtained by moving the camera over the samples. Since there were a total of 81 samples, a total of nine hyperspectral image acquisitions were made. All samples were subjected to an optical stabilization process where they were left in the experimental environment for 20 min prior to hyperspectral data acquisition. This was executed to eliminate any coincidental differences that might be caused by differences in myoglobin/oxymyoglobin content, and to ensure that the samples were as constant as possible before proceeding with spectral analyses [32]. This process ensures that the acquired hyperspectral images accurately reflect the characteristics of the samples under investigation. The use of this specialized imaging

system, combined with experimental protocols, resulted in a high-quality dataset suitable for subsequent analysis and interpretation.

The process of separating the red meat part from the entire sample image was performed. In the hyperspectral data cube, the red meat was set as the region of interest (ROI) and spectra were extracted from that region only. The hyperspectral imaging device used in this study had a measurement wavelength band of 400–1000 nm and a spectral resolution of 2 nm, so 300 spectral images are acquired in one acquisition to form a hyperspectral data cube. These 300 single-wavelength images have a pixel resolution of 700 × 900 per image, and to extract only the red meat from each spectral image, ten images corresponding to the red (630–650 nm) and green (540–560 nm) wavelength bands were extracted separately [33]. Figure 3 is an image reconstructed by selecting only the data in the appropriate wavelength band from the hyperspectral data cube in which samples from each experimental group were taken. The mean was estimated for each of the ten images extracted by each domain to create a single image, from which a contrast image of 50% relative intensity as the threshold was produced. The contrast images of red and green domains were super-imposed, and after subtraction, the area activated only in the red domain was determined. This process ensures that only the spectrum of the red meat is extracted for analysis from a hyperspectral data cube that contains spectra of multiple components. The extracted spectra were either used for model development without preprocessing or preprocessed to create new datasets. Four preprocessing techniques were used for model development. These were multiplicative scatter correction (MSC), standard normal variate (SNV) transformation, first-order Savitzky–Golay filtering, and min-max normalization.

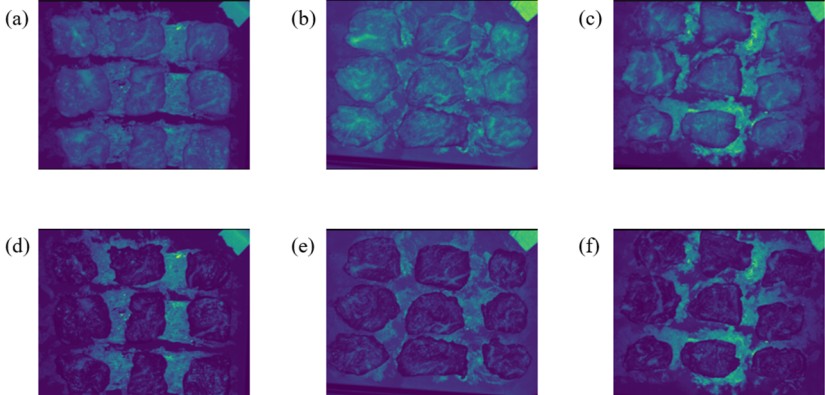

**Figure 3.** Spectral images in the hyperspectral data cube acquired in this study. (**a**–**c**) 630–650 nm average image of 'Fresh', 'Abused', and 'Frozen' samples, respectively; (**d**–**f**) 540–560 nm average image of 'Fresh', 'Abused', and 'Frozen' samples, respectively.

### 2.4. Construction of Classification Models

PLS-DA and SVM were used to construct a classification model. PLS was used to locate the optimal regression vector between the X matrix as the data to be analyzed and the Y vector as the outcome. Unlike PLS, PLS-DA applies dummy variables as class indicators instead of estimations of the Y vector [34]. In this study, the X matrix was the spectrum and the Y vector was the experimental class to which the spectrum belonged. A simple binary classification was impossible as the model was required to discriminate between three experimental classes, so three-dimensional eigenvectors were used. The eigenvectors in this study were defined as follows: (1, 0, 0) for the 'Fresh' class; (0, 1, 0) for the 'Abused' class; and (0, 0, 1) for the 'Frozen' class. The PLS-DA applied after defining the eigenvector of each class produced the result in a vector form ($x_i$, $y_i$, $z_i$). As the sum of all result vectors is generally 1, the outcome regarding which experimental class the sample belongs to can be determined based on whether the direction of the vector is >0.5 [35]. For instance, when the output of PLS-DA is (0.7, 0.25, 0.05), the x-axis vector is 0.7, which is >0.5, indicating

that the spectrum is of the 'Fresh' class based on the x-axis eigenvector. Equation (2) is the general expression of PLS-DA [36]:

$$Y = Xb + E,\qquad(2)$$

where X matrix is the spectral data, Y vector is the measured reference, b is the regression coefficient, and E is the error term.

The SVM was used as a supervised ML classifier. It is a hyperplane-based method of binary classification. Hence, for it to be applied in multi-class classification, a higher number of computations than those used in the binary classification of the same number of datasets is required, as the number of variables increases proportionately to the number of classes [37]. The decision functions for additional support included 'one-against-all', 'one-against-one', 'all-together', and 'direct acyclic graph' [38]. This study used Python version 3.8.5 (Python Software Foundation, Wilmington, DL, USA) for SVM computations. The SVM module in Python provides the decision functions 'one-against-all' and 'one-against-one.' In 'one-against-all', a hyperplane is generated for each class to differentiate one from all the rest. In 'one-against-one', all hyperplanes that identify two random classes are generated. As the unselected classes are excluded, the number of datasets in each hyperplane computation is reduced, allowing the results to be rapidly generated. In this study, the 'one-against-one' function was selected for all SVM models. The number of hyperplanes generated by the two decision functions was identical, with insignificant variation in the computation speed, as the models aimed at isolating three classes. Nevertheless, the 'one-against-one' function was more effective than the 'one-against-all' function, considering the characteristic use of meat samples, the quality of which gradually falls from fresh meat rather than the samples of entirely different qualities.

The spectra used in this study were multi-dimensional datasets with 300 inputs, which implied that the hyperplanes might need to be more easily separated. One of the ways to achieve mapping without linear separation of the data is to use kernel functions [39]. Four of the kernel functions frequently used in the SVM were applied in this study, i.e., linear, polynomial, radial basis function (RBF), and sigmoid, and each of the equations are shown in Equations (3)–(6), respectively:

$$k(x, y) = x^T y + c,\qquad(3)$$

$$k(x, y) = \left(x^T y + c\right)^3,\qquad(4)$$

$$k(x, y) = \exp\left(||x - y||^2\right),\qquad(5)$$

$$k(x, y) = \tanh\left(x^T y + \theta\right),\qquad(6)$$

where the X matrix is the spectrum and the Y vector is the experimental class.

The F1 score and accuracy of the 'Fresh' class were used for the classification model performance indicators. Accuracy is the most commonly applied performance indicator for classification models. It ensures the most intuitive and rapid indication of performance by defining the proportion of correct outputs among the total number of classified spectra. The F1 score is the harmonic mean of precision and recall calculated for each class. In bi-nary classification, precision and recall, which correspond to sensitivity and specificity, could be vastly different values, depending on the situation [40]. Hence, the harmonic mean was estimated for the two scores to mutually complement the F1 score. In a multi-class classification model, the F1 score for the class is a single-digit representation of whether the prediction on a specific class is valid [41]. The respective accuracy and F1 score of the 'Fresh' group calculations are shown in Figure 4.

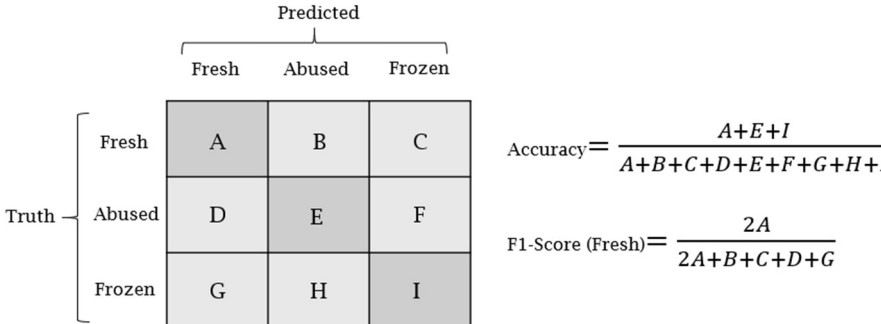

**Figure 4.** Calculation of two indicators (accuracy and F1 score for a particular class) to evaluate the performance of a model from a confusion matrix.

## 3. Results

### 3.1. Spectrum Extraction

After applying the spectral extraction process using ROI selection, 4950 hyperspectral spectra were finally selected from the hyperspectral data cubes acquired with the hyperspectral imaging system. A graphical representation of the entire spectrum is shown in Figure 5a, and Figure 5b shows the mean and standard deviation of the spectrum for each class. To construct a ML model, it is essential to use data that have not been used for model training to evaluate its performance. This investigation randomly divided the experimental data into a training set and a test set in an 80:20 ratio. The training set, which was randomly selected, was subjected to a 10-fold cross-validation process to validate the performance and to develop an optimal model. The test set was also used to evaluate the performance of the models, and the confusion matrix of all models shows the result of the evaluation.

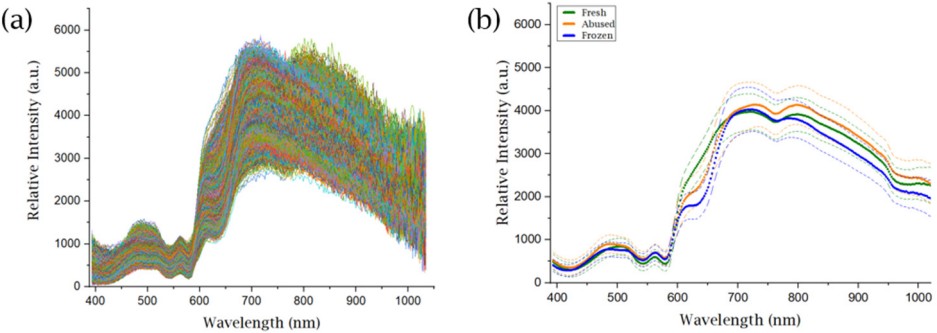

**Figure 5.** Spectra from the hyperspectral data cube which were used to develop classification models of meat samples. (**a**) Full spectra of experimental data; (**b**) mean spectrum of each experimental group (solid line) and with added and subtracted standard deviations (dashed line).

### 3.2. Drip Loss Test

Figure 6 shows a box chart of the results of the drip loss test according to the sample treatment. There was a clear distinction among the three interquartile boxes for the 'Fresh', 'Abused', and 'Frozen' classes, while an outlier was detected only in the 'Abused' class. For fresh beef, the drip loss was ≤1%, indicating almost no drip loss upon cooking. For the beef that had been refrigerated for 8–15 d, the drip loss was 1–1.48%, and in one test, an outlier of 2.662% was detected. For the beef that had been frozen for 6 d and thawed, a drip loss of up to 8% was observed. It is well known that freezing and thawing meat increases the drip loss due to changes in the cell structure and the water holding capacity of the meat [42]. When meat is frozen, ice crystals form within the meat, which can damage the cell walls and cause them to rupture. This damage can result in a loss of water-soluble compounds and an increase in free water in the meat. We can also see that prolonged refrigeration of meat also affects the drip loss. This is probably because prolonged refrigeration can also cause protein denaturation and enzymatic degradation of the meat. However, as the results

show, the effect of refrigeration on drip loss is small compared to freezing. Additionally, this result is similar to that found in other studies [43].

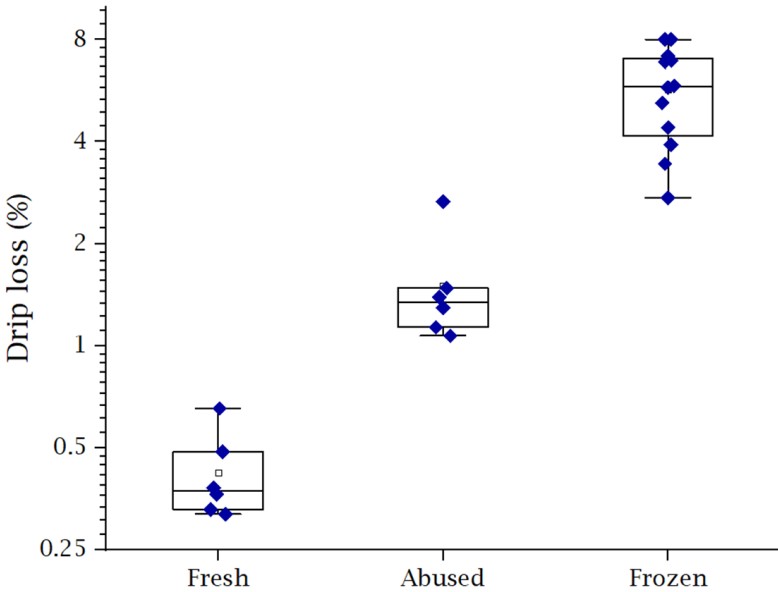

**Figure 6.** Box plot comparing drip loss experiment results among different treatments of beef samples, showing quartiles and outliers.

Table 1 presents the results of statistical analyses on the collected data. The results of the one-way analysis of variance confirmed significant differences across the experimental classes with an F-value of 39.572 and a significance probability of 0.000. Using Scheffe's post hoc test, the differences between each variable were analyzed, and the statistical significance for the 'Fresh' and 'Abused' classes was found to be 0.381, a level above the criteria at 0.05, indicating no significant differences across the data of 'Fresh' and 'Abused' classes. The significance level for the 'Frozen' class and the other two classes was 0.000, indicating a clear distinction.

**Table 1.** One-way ANOVA results for drip loss experiment.

| Class | M ± S.D. | F Value for ANOVA | Scheffe Test ($p < 0.05$) |
|---|---|---|---|
| Fresh (a) | 0.423 ± 0.128 | | |
| Abused (b) | 1.505 ± 0.487 | 39.572 | ab < c |
| Frozen (c) | 5.690 ± 1.776 | | |

*3.3. Performance of Classification Model*

Classification models using the PLS-DA to classify the 'Fresh', 'Abused', and 'Frozen' classes were constructed and the model performance was evaluated. For the NIR spectra, four preprocessing methods were applied, which included the multiplicative scatter correction (MSC) and the standard normal variate (SNV) for correction, the Savitzky–Golay 1st filter for smoothing, and the Min–Max for normalization. A model was constructed for the data in each preprocessing step as well as for the raw data, which led to a total of five models. The model performance was evaluated as follows: first, a 10-fold cross-validation was performed on the training set, and for the model demonstrating the highest accuracy, the test set was applied to determine the final performance. For the classification result on the test set, the F1 score as well as the accuracy were estimated to verify whether the model could accurately classify the 'Fresh' class even if it could not differentiate between 'Abused' and 'Frozen'.

Table 2 presents the evaluation results. First, the accuracy from the training set was ≥94% for all the preprocessing methods, suggesting that the learning through cross-validation led to discrimination among the experimental classes. Similarly, the accuracy in applying each model to the test set was 92.71–95.33%, suggesting a reliable level of performance for field applications. Next, considering the F1 scores of each model, the F1 score of the 'Fresh' class was higher than the other two classes in all cases. This suggests that most of the unclear distinctions that the models are confused about are between the 'Abused' and 'Frozen' classes. Although there are significant differences in the drip loss values, the analysis of the spectra extracted from the hyperspectral data cube shows that there are similarities between the meat samples that have been refrigerated for a long time and those that have been frozen. Both the accuracy and F1 score implied that the model most capable of differentiating between fresh meat and non-fresh meat was the one constructed using the SNV preprocessing data.

**Table 2.** Performance of PLS-DA classification models for each group of samples according to the preprocessing methods (unit: %).

| Preprocessing | Training Accuracy | Test Accuracy | Test F1 Score | | |
|---|---|---|---|---|---|
| | | | Fresh | Abused | Frozen |
| No preprocess | 95.18 | 94.37 | 99.28 | 90.58 | 92.28 |
| MSC | 94.84 | 94.60 | 98.91 | 90.16 | 93.25 |
| SNV | 95.87 | 95.66 | 99.44 | 92.76 | 94.13 |
| Savitzky–Golay 1st | 95.06 | 93.47 | 99.03 | 88.92 | 90.97 |
| Min-Max | 94.91 | 92.71 | 98.69 | 87.90 | 90.48 |

The classification models for SVM were constructed in the same way as those for PLS-DA. As with the PLS-DA models, the training data were categorized into five classes according to the preprocessing method, and a total of 20 SVM models were constructed using four kernel functions for data application. As mentioned in the 'Materials and Methods' section, the four kernel functions were linear, polynomial, radial basis function (RBF), and sigmoid. The method of model performance evaluation was also identical to that previously described. After the 10-fold cross-validation on the training set, the model performance was evaluated based solely on the accuracy, and when the accuracy was <50%, the training was considered a failure and the test set was not applied. For the model showing a level of accuracy above the training criteria, the test set was applied to evaluate the performance. For the result of the test set, the accuracy and the F1 score of each class were estimated to independently verify the ability to identify fresh meat.

Table 3 presents the evaluation results for the performance of all 20 models according to the combination of kernel functions and preprocessing methods. As previously stated, three models did not satisfy the training criteria of accuracy, all of which used the sigmoid kernel function. This suggested that the model performance could vary greatly according to the state of preprocessing and the use of the kernel function. Among the 17 remaining classification models, 14 models showed a level of accuracy of ≥83% and three models classified all spectra of the test set in the 'Fresh' class, showing a 33.33% accuracy. The accuracy on the training set was 100% for these models, suggesting that overfitting occurred in the process of hyperplane generation. Additionally, among the 20 cases created by the combination of preprocessing methods and kernel functions, the SNV preprocessing and RBF kernel function resulted in the highest F1 score for the 'Fresh' class. The accuracy was 96.57%, with a 100% classification of the 'Fresh' class. However, as overfitting occurred in the other three models applying the RBF kernel function, the model performance should be further validated. The overall performance evaluation on the kernel function implied that the models applying the polynomial kernel function could achieve a relatively stable performance. Additionally, unlike the PLS-DA models, the F1 scores of the 'Abused' and 'Frozen' classes also differed, with the 'Abused' class being more difficult to distinguish

overall. It is speculated that the fresh and frozen meat samples have unique spectral features, while the meat samples that have been refrigerated for a long time have a mixture of features from the other two experimental groups.

**Table 3.** Performance of SVM classification models for each group of samples according to the combination of kernel functions and preprocessing methods (unit: %).

| Preprocessing | Kernel Function | Training Accuracy | Test Accuracy | Test F1 Score | | |
|---|---|---|---|---|---|---|
| | | | | Fresh | Abused | Frozen |
| No preprocessing | Linear | 95.51 | 90.30 | 98.67 | 85.00 | 86.92 |
| | Polynomial | 93.99 | 93.13 | 99.55 | 88.40 | 90.98 |
| | RBF | 100 | 33.33 | 50.00 | 0.00 | 0.00 |
| | Sigmoid | 28.86 | - | - | - | - |
| MSC | Linear | 95.05 | 90.00 | 98.40 | 85.51 | 85.53 |
| | Polynomial | 91.74 | 90.20 | 98.55 | 84.68 | 86.80 |
| | RBF | 100 | 33.33 | 50.00 | 0.00 | 0.00 |
| | Sigmoid | 85.73 | 83.84 | 95.38 | 72.22 | 81.57 |
| SNV | Linear | 96.54 | 94.44 | 99.59 | 91.11 | 91.85 |
| | Polynomial | 93.01 | 91.62 | 99.59 | 85.81 | 88.02 |
| | RBF | 99.97 | 96.57 | 100 | 94.10 | 95.03 |
| | Sigmoid | 84.97 | 84.65 | 96.40 | 72.36 | 81.59 |
| Savitzky–Golay 1st | Linear | 95.33 | 90.81 | 99.27 | 84.55 | 87.82 |
| | Polynomial | 93.48 | 93.74 | 99.42 | 88.93 | 92.10 |
| | RBF | 100 | 33.33 | 50.00 | 0.00 | 0.00 |
| | Sigmoid | 29.32 | - | - | - | - |
| Min-Max | Linear | 94.85 | 93.61 | 99.79 | 91.68 | 92.53 |
| | Polynomial | 98.48 | 96.97 | 99.85 | 95.18 | 95.69 |
| | RBF | 99.39 | 97.68 | 99.71 | 96.43 | 96.76 |
| | Sigmoid | 30.58 | - | - | - | - |

*3.4. Confusion Matrix*

The results of classification model construction can be visualized through a confusion matrix. Tables 4 and 5 show the confusion matrix of the highest performance models developed with PLS-DA and SVM, respectively. Since the total data consist of 4950 spectra and the test set consists of 990 spectra (or 20% of them), the sum of all elements in the confusion matrix is 990. Comparing the two tables, the results show that the classification results are very close in percentage despite the models being developed with completely different methods. Looking at the detailed prediction results, the top row and the left column are correctly classified as they are basically the result of models with high F1 scores in the 'Fresh' class. This suggests that the classification model is able to distinguish between fresh and not fresh meat. The distinction that all models had relative difficulty classifying was between the 'Abused' and 'Frozen' classes. This indicates that when attempting to predict the condition of meat through spectral analyses, it is reasonable to assume that when a sample was not frozen, but kept in refrigeration for a long period of time, it deteriorates and becomes similar in quality to a frozen sample. Additionally, considering the overall accuracy, most of the models that discriminate 'Fresh', 'Abused', and 'Frozen' classes by ML techniques using near-infrared hyperspectral images of beef samples succeeded in discriminating more than 90%; thus, it can be said that it is feasible to apply them in actual meat purchasing sites in the future. However, due to the nature of biological materials, the degradation process during the chilling and freezing process may be different from one carcass to another, so further research is needed.

**Table 4.** Confusion matrix of test results for the PLS-DA model which showed the highest performance indicators.

| True Class | Predicted Class | | |
|---|---|---|---|
| | **Fresh** | **Abused** | **Frozen** |
| Fresh | 357 (99.2%) | 2 (0.5%) | 1 (0.3%) |
| Abused | 1 (0.3%) | 269 (89.7%) | 30 (10.0%) |
| Frozen | 0 (0.0%) | 9 (2.7%) | 321 (97.3%) |

**Table 5.** Confusion matrix of test result for the SVM model which showed the highest performance indicators.

| True Class | Predicted Class | | |
|---|---|---|---|
| | **Fresh** | **Abused** | **Frozen** |
| Fresh | 360 (100%) | 0 (0%) | 0 (0%) |
| Abused | 0 (0%) | 271 (90.3%) | 29 (9.7%) |
| Frozen | 0 (0%) | 5 (1.5%) | 325 (98.5%) |

## 4. Conclusions

An NIR hyperspectral image acquisition device was used to construct models to discriminate between the 'Fresh', 'Abused', and 'Frozen' states of beef in a non-invasive way. The samples of beef produced in South Korea (the test material) were divided into freshly refrigerated (12–24 h), long-term refrigerated (8–15 d), and thawed states. In all experiments, nine hyperspectral image data cubes were obtained and drip loss tests were performed to quantitatively analyze the condition of the meat samples. In this study, 4950 spectra were obtained, of which 80% was used as a training set and 20% as a test set. With four preprocessing methods, which included MSC and SNV for correction, Savitzky–Golay 1st filter for smoothing, and Min–Max for normalization, and the raw data, a total of five datasets were prepared for ML model construction. The PLS-DA and SVM techniques were used to construct the models, with four kernel functions in the case of SVM models. Accuracy was the primary performance indicator and the F1 score on the 'Fresh' class was additionally estimated to independently validate the performance of fresh meat classification. The performance of the test set on the basis of accuracy was more than 90% for almost all of the models developed, and most errors were due to its failure to differentiate between the 'Abused' and 'Frozen' classes. The performance was the highest for the SVM model applying scatter correction and the RBF kernel function (accuracy = 96.57% and F1 (Fresh) = 100%). The results of this study indicate that a prediction model to distinguish fresh meat from non-fresh meat can be constructed using the spectra screened by purifying hyperspectral image data cubes.

**Author Contributions:** Conceptualization, S.P. and G.K.; methodology, S.P., S.-J.H. and G.K.; software, S.-J.H. and S.K.; validation, S.K., J.R. and S.R.; formal analysis, S.P. and S.K.; investigation, J.R. and S.R.; resources, S.P.; data curation, J.R. and S.R.; writing—original draft preparation, S.P.; writing—review and editing, G.K.; visualization, S.P. and J.R.; supervision, G.K.; project administration, G.K.; funding acquisition, G.K. All authors have read and agreed to the published version of the manuscript.

**Funding:** This work was supported by the Korea Institute of Planning and Evaluation for Technology in Food, Agriculture and Forestry (IPET) through the Smart Agri Products Flow Storage Technology Development Program (no. 1545027072). It was funded by the Ministry of Agriculture, Food and Rural Affairs (MAFRA), the Cooperative Research Program for Agriculture Science and Technology Development (no. PJ017095), and the Rural Development Administration, Republic of Korea.

**Institutional Review Board Statement:** Not applicable.

**Data Availability Statement:** The data presented in this study are available upon request from the corresponding author. Data are not publicly available due to the funding agency policy.

**Acknowledgments:** We gratefully acknowledge the technical support of Cheorun Jo and Dong Gyun Yim from the Animal Origin Food Science Laboratory of Seoul National University.

**Conflicts of Interest:** The authors declare no conflict of interest. The funders had no role in the design of the study; in the collection, analyses, or interpretation of data; in the writing of the manuscript; or in the decision to publish the results.

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
