# Peer review of "Classification of Fresh and Frozen-Thawed Beef Using a Hyperspectral Imaging Sensor and Machine Learning"

_agriculture, doi:10.3390/agriculture13040918_

Round 1

Reviewer 1 Report

The authors present a novel technique for categorizing beef into three distinct states: fresh, frozen, and deteriorated or abused. The authors employ hyperspectral data from the VIS-NIR region (400-1000 nm). Their proposal is based on partial least squares discriminant analysis and support vector machines. The labels used in the study are substantiated by experimental data from a drip-loss test. The classification approach was evaluated using four distinct pre-processing methods, as well as no pre-processing and four different SVM kernels. Ten-fold cross-validation was employed in the simulations.

The findings are interesting, and the optimal configuration demonstrates that the model exhibits great performance in distinguishing fresh beef from non-fresh samples. Overall, the manuscript is well-written and easily comprehensible. I kindly request the authors to address the following minor comments:

1.       In Section 2.4, the authors discuss their classification model (lines 179-193) and describe the variables associated with input data and labels. However, the presentation would benefit from the inclusion of relevant mathematical equations to illustrate the models and variable sizes. While a paper may not be entirely self-contained, providing these equations would make it more accessible to readers who are not experts in machine learning techniques.

2.       Although the paper may have length constraints, it would be valuable for readers to understand the performance of other non-optimal configurations from Table 3 in classifying non-fresh (frozen and thawed) samples. How does the performance of these configurations compare to the SNV/RBF approach?

3.       To enhance readability and comprehension, it would be helpful to present the confusion matrix in Table 4 as percentages rather than raw numbers. This would provide a more intuitive representation of the data for readers.

Author Response

Please see the attachment. Thanks for your valuable comments.

Reviewer 2 Report

This manuscript (agriculture-2334530) develops a machine learning model using hyperspectral image data to differentiate between fresh and non-fresh beef meat, offering a rapid and non-invasive method for routine meat storage state analysis.

The introduction, materials and methods, and conclusion sections are satisfactory. However, the article lacks a discussion of the results. Additionally, the experimental design appears peculiar, as it does not follow the same periods/time of analyses as shown in Figure 1. Could you please explain why the time periods differ between 7 days and 6 days?

It is interesting, but requires some changes according to the following comments:

Keyword in alphabetic order;

In each of the figure captions, add the number of repetitions when necessary. Include information so that we can view the image and interpret it alongside the caption, without needing to refer back to the text. This allows for a more fluid reading experience for the readers.

Did you try conducting tests with other algorithms or classification systems, such as random forest, neural networks, AdaBoost, etc?

Why can Savitzky-Golay be used for classification, or is it only suitable for smoothing/correction?

Please review the references again and update them as necessary. Some of them are outdated.

Best regards

Author Response

(The authors gave the same response as above.)

Round 2

Reviewer 2 Report

I consider that the authors have made the necessary improvements to the manuscript, and I recommend it for publication.